# Consumer Preference of Traditional Korean Soy Sauce (*Ganjang*) and Its Relationship with Sensory Attributes and Physicochemical Properties

**DOI:** 10.3390/foods12122361

**Published:** 2023-06-13

**Authors:** Yang Soo Byeon, JeongAe Heo, Kwon Park, Young-Wook Chin, Sang-pil Hong, Sang-Dong Lim, Sang Sook Kim

**Affiliations:** 1Research Group of Food Processing, Korea Food Research Institute, Wanju-gun 55365, Republic of Korea; b.yangsoo@kfri.re.kr (Y.S.B.); heo.jeongae@kfri.re.kr (J.H.); parkgwon@kfri.re.kr (K.P.); 2Traditional Food Research Group, Korea Food Research Institute, Wanju-gun 55365, Republic of Korea; ywchin@kfri.re.kr (Y.-W.C.); sphong@kfri.re.kr (S.-p.H.); limsd@kfri.re.kr (S.-D.L.)

**Keywords:** *ganjang*, free amino acids, organic acids, preference mapping, consumer

## Abstract

This study aimed to investigate the physicochemical characteristics, sensory attributes, and consumer acceptance of the Certification of Quality of Traditional Food (CQT) *ganjang* samples produced in different provinces of Korea. Wide variations in physicochemical properties were found among the samples, especially in lipids, total nitrogen, acidity, and reducing sugar. Traditional fermented foods are known to be closely tied to regional features, but the composition and characteristics of CQT *ganjangs* might be influenced much more by individual *ganjang* producers than by region. Preference mapping was performed to understand consumer behavior towards *ganjang*, and most consumers tended to have similar preferences, implying shared a common sensory ideal. The results of the partial least squares regression revealed drivers of liking for *ganjang* among sensory attributes, free amino acids, and organic acids. Overall, sensory attributes such as sweetness and umami were positively associated with acceptability, while the terms related to fermentation were negatively associated. In addition, amino acids, such as threonine, serine, proline, glutamate, aspartate, and lysine, and organic acids, such as lactate and malate, were positively associated with consumer acceptance. The important implications of the findings of this study for the food industry can be utilized to develop and optimize traditional foods.

## 1. Introduction

Traditional Korean soy sauce, called *ganjang*, has a long history in Korea that can be traced back to the 7th century AD [1]. It is made from only *meju* (fermentation bricks made of soybeans containing wild-type microbiota), salt, and water [2], and is fermented spontaneously without pitching other starters [3]. The filtrate is then separated and matured, which gives it a complex flavor and mouthfeel profiles of salty, slightly sweet, thick, and deeply savory. Indeed, *ganjang* is used to add flavor to a variety of dishes, including soups, stews, marinades, dipping sauces, and stir fries, as a staple condiment used in Korean cuisine.

It is not only a flavoring sauce; *ganjang* also has functional aspects. Researchers have reported health benefits, such as the inhibition of type 2 diabetes and cancers, as well as inflammatory and oxidative stress [4,5,6,7]. Thus, with the increased popularity of emerging healthy foods, there has been growing interest in traditional fermented foods [8]. Worldwide, there are certification systems for traditional foods to sustain inherent traditional foods. For example, the Traditional Speciality Guaranteed established by the EU [9] covers traditional recipes and production methods. Similarly, several countries have adopted a geographical indication system to protect food products in specific regions, which usually include traditional foods [10]. These certification systems, along with many others worldwide, play an important role in protecting and promoting traditional food and food cultures.

In Korea, an institutional quality certification system for traditional food, the Certification of Quality of Traditional Food (CQT), is in place and administered by the Ministry of Agriculture, Food, and Rural Affairs [11]. The CQT aims to encourage production and provide consumers with high-quality authentic products. It is awarded to traditional food products that meet a certain CQT standard. *Ganjang* is one of the main traditional food products eligible for certification; in 2023, 91 producers registered in a category called T016. Recent research on ganjang has mostly concentrated on profiling metabolites and microbial communities [12,13]. Regarding consumer studies, there are effects of consumer liking on carriers and testing conditions [14,15], sensory profiling of different types [16], and maturation periods [17], but little information is available on the taste of CQT *ganjang*, its regional characteristics, and underlying consumer responses.

Preference mapping is a valuable tool for portfolio management that allows researchers to visualize how products’ different attributes or compositional profiles affect consumer preference [18,19]. This involves gathering data from consumers on their perceptions of various products and analyzing them using multivariate analysis techniques. Preference maps provide a clear and concise representation of data, allowing researchers to easily identify patterns and trends [20,21]. These approaches have been implemented in studying a variety of products, such as strawberry vinegar [22], chocolate milk [23], lager beer [24], soybean paste (*doenjang*) [25], coffee [21,26,27], and snacks [20,28]. However, the studies rarely identified preference mapping of *ganjang* and other soy sauce products.

The objective of this study was to provide insights into CQT *ganjang* by focusing on flavor-related components and its consumer response and to promote consumer acceptance; such products could potentially guide improvements in product quality. In this study, consumer acceptance, sensory attributes, physicochemical properties, and their relationships were investigated in representative *ganjang* samples from different provinces.

## 2. Materials and Methods

### 2.1. Ganjang Samples

Thirty-six CQT *ganjang* products were used in this study. These were representative QCT *ganjang* samples from various provinces in Korea: three from Gangwon (GW), five from Gyeonggi (GG), nine from Chungcheong (CC), eight from Jeolla (JL), eight from Gyeongsang (GS), and three from Jeju (JJ) (Figure 1a). The samples were selected from the products of 91 certificated manufacturers, considering the business size and sales volume of each region. While the conventional production of CQT *ganjang* is strictly required for the use of only *meju* as a starch source, it is permissible to make extensive use of other additives during the fermentation process within the bounds of accepted tradition. Thus, some CQT *ganjang* products also include a wide range of additives, such as meats, seafood, and different types of wood and berries (Figure 1b).

### 2.2. Physicochemical Properties

#### 2.2.1. Proximate Analysis, Color, pH, Acidity, Salinity, and Reducing Sugar

Proximate analyses were conducted on total solid content (TS), crude ash, lipid, and total nitrogen (TN), according to the Association of Official Analytical Chemists (AOAC) methods: 925.10, 945.28, 991.36, and 920.53 [29], respectively, expressed as percentage as weight per weight (% *w*/*w*). The color was measured using an L (lightness), a (redness), and b (yellowness), c (chroma), and h (hue) value with the color hunter system of a CM-5 spectrophotometer (Konica Minolta, Tokyo, Japan). The pH was measured using a 720A pH meter (Orion Research Inc., Boston, MA, USA).

Total acidity was measured according to the titrimetric method of AOAC 973.42. Total acidity (TA) was determined by the amount of 0.1 N sodium hydroxide to an endpoint of pH 8.2 in the sample and expressed as g/100 mL lactic acid equivalent. Salt content (g/100 mL) was measured by the Mohr method, as described by Belcher et al. [30]. Reducing sugar (RS) content was determined by the 3,5-dinitrosalicylic acid method [31], with some modifications to add a clarification step to avoid colorimetric interference due to the dark brown color in the samples. Briefly, each sample was properly diluted in distilled water, mixed with 5% magnesium oxide (*w*/*v*), and shaken overnight. The clarified solution was then filtered and used to determine the RS. Absorbance at 550 nm was measured using a microplate reader (SpectraMax i3, Molecular Devices, Sunnyvale, CA, USA). The results were expressed in milligrams of glucose equivalents per 100 milliliters (mg/100 mL). All tests were performed in triplicates.

#### 2.2.2. Free Amino Acids

Free amino acids (FAAs) in *ganjang* samples were analyzed using the Korean Food Standards Codex 2.1.3.3 [32], with some modifications. A high-speed amino acid analyzer, L-8800 (Hitachi High-Tech Co., Tokyo, Japan), was attached to an ion exchange column #2622SC PF 4.6 mm i.d. x 60 mm (Hitachi High-Tech Co., Tokyo, Japan). The mobile phase used a PF1, PF2, PF3, PF4, PF-RG, R-3, C-1, ninhydrin solution, and a buffer solution (Wako Pure Chemical Industries, Osaka, Japan). Two milliliters of the sample were extracted in a rotary shaker for 30 min with 100 mL of 16% trichloroacetic acid and then centrifuged at 3000 rpm for 15 min at 4 °C. Each supernatant-obtained extract was filtered with a 0.2 μm syringe filter (Life Science, Boston, MA, USA). The filtrated samples were injected into the analyzer system with a postcolumn for ninhydrin derivatization. As internal standards, a mixture solution was prepared of 1:1 (*v*/*v*) of type AN-II #015-14461 and B #016-08641 (Wako Pure Chemical Industries, Osaka, Japan) with various concentrations. The calculation of FAA content was conducted by systematically comparing them with relevant standards using an EZChrom Elite software. The measurements were repeated twice and represented as the averages.

#### 2.2.3. Organic Acids

Organic acids (OAs) in *ganjang* samples were quantified by a high-performance liquid chromatography (HPLC) (Agilent 1260 Infinity, Memphis, TN, USA) equipped with the Aminex HPX-87H Column (300 × 7.8 mm, BioRad Laboratiories, Inc., Hercules, CA, USA) and diode array detector at 210 nm. The column heated at 50 °C was used to analyze 20 μL of properly diluted *ganjang* samples. Sulfuric acid solution (0.008 N) was used as the mobile phase at a flow rate of 0.6 mL/min. The standard response curve for OAs was a linear regression fitted to values obtained at each of the six concentrations, 0–100 mg/L, with high correlation coefficients (R^2^ > 0.99). Quantification of organic acid content was calculated based on the peak area in accordance with identical retention times of standard resources. The measurements were performed in triplicate; the relative standard deviation values were less than 1.0% for the peak areas.

### 2.3. Sensory Analysis

The study participants were recruited from a pool of panelists at the Sensory Service Center at the Korea Food Research Institute (Wanju, Republic of Korea). The selection criteria for the participants were based on their interest in traditional foods and frequency of consuming *ganjang* products (consumed twice or more within the last month). To ensure reliable statistical generalizability and validation of the study findings, a minimum of 100 subjects were invited. In total, 101 subjects (37 men and 64 women) in the range of 20–60 years old participated in this study. Informed consent was obtained from each participant before they filled out the questionnaire, and all participants confirmed that they had no allergies to soybeans and were willing to participate in the study. Sensory evaluations were conducted in individual sensory booths, and data were recorded directly on a computerized data collection system (Compusense Inc., Guelph, ON, Canada). As a sample presentation protocol, each *ganjang* sample (10 g) was presented in a 2 oz transparent cup covered with a lid, and a 0.2 mL teaspoon was also provided. Each sample was evaluated for 10 min, with a five-minute wait between samples to prevent assessors’ fatigue and allow sensory receptors to recover. Water and unsalted crackers were provided as a palette cleanser between the samples. Nine samples were evaluated per session, and each test session was conducted over four consecutive days. To avoid any potential bias, the samples were presented according to the Williams Latin Square design [33]. Monetary compensation was given to the participants. All sensory study adhered to the ethical principles outlined in the Declaration of Helsinki [34]. Necessary approval was granted by the Ethical Committees of the Institutional Review Board of the Korea Food Research Institute (approval code: KFRI 2022-09-002-001).

For the evaluation of *ganjang* samples, participants were asked to evaluate the acceptability of each sample using a nine-point hedonic scale for appearance, aroma, taste/flavor, mouthfeel, and overall acceptability. Subsequently, the magnitudes of sensory attributes were evaluated using an eight-point category scale, including ‘none’, with zero points referring to not perceived at all. All questions were presented in a balanced order within each category, as recommended by Ares et al. [35]. To determine the intensity of sensory attributes, the questionnaire consisted of a list of terms that were predefined by prior sensory studies on soy sauce [18,36,37,38,39,40], which had studies focusing on sensory properties. In total, the 36 sensory attributes included an appearance (color), 13 odors (pungent, alcohol, briny, burnt, fermented, fermented fish, roasted soybean, beany, sour, sweet, dusty, *meju*, grain, and chemical), 13 tastes/flavors (sweetness, sourness, saltiness, bitterness, umami, chemical, alcohol, fermented, fermented fish, *meju*, roasted soybean, beany, and burnt), 4 mouthfeels (astringent, metallic, biting, and body), and 5 aftertastes (sweet, sour, salty, bitter, and umami).

### 2.4. Statistical Analysis

For the physicochemical properties and consumer data, analysis of variance (ANOVA) was performed to determine the differences among the *ganjang* samples. When a difference among samples was found, significant differences among samples were calculated using the Student–Newman–Keuls (SNK) multiple comparison test at *p* < 0.05. In internal preference mapping (IPM), for an overview of the consumer preference segmentations, the overall acceptance score of whole consumers was dimensionally reduced using PCA to create an internal preference space. Next, hierarchical clustering on principal components (HCPC) was conducted using Ward’s method of agglomeration and Euclidean distances [41]. To compare each cluster, statistical analysis was conducted using one-way ANOVA with Fisher’s LSD multiple comparison test as a post hoc analysis; the confidence level for the analysis was set at 95%. Moreover, to capture the overall trend in consumer preference, an external preference map (EPM) was illustrated based on PCA perceptual space by sensory matrix, and then applied to locally weighted regression smoothing (LOESS) with generalized additive models (GAMs) to consider both linear and nonlinear relationships between variables [42]. They were estimated using LOESS, which fits a separate regression line to each data point, with the regression weighted toward nearby points. The goodness of fit of the GAM in these data was checked a StabMap function in SensMap followed by the guidance of Rebhi and Malouche. The stability indicators were 14.7 and 15.4, calculated by StabMap; measuring the distance between estimates and actuals was the lowest, implying how well the visualized maps performed on the whole sample [43]. Finally, partial least squares regression (PLS-R) analysis was carried out to determine the drivers of liking *ganjang* products. As explanatory variables (Xs), physical properties, chemical compositions, and sensory attributes were used. The consumer acceptance data were applied as dependent variables (Y). All statistical analyses were performed using Xlstat statistical software ver. 2022.2.1. (Addinsoft, Paris, France) or R packages of FactoMineR [44], factoextra [45], and SensMap [43] in R languages [46].

## 3. Results

### 3.1. Physicochemical Properties

#### 3.1.1. Proximate Analysis, Color, pH, Acidity, Salinity, and Reducing Sugar

TS, ash, lipid, TN, color-L, a, b, c, and h, pH, acidity, salinity, and RS values of the ganjang samples are shown in Table 1. In the proximate analysis, TS content was in the range of 23.8% (JL8) to 54.9% (CC5), with an average of 33.6%. The ash content ranged from 15.6% (JL8) to 26.6% (CC2), with an average of 21.8%. The lipid contents were in the range of 0.08% (JL1) to 0.81% (JL7), with an average of 0.41%. The TN content ranged from 0.26% (CC2) to 2.40% (GG4), with an average of 0.97%. The pH ranged from 4.78 (JL7) to 6.88 (CC2), with an average of 5.48. The acidity content ranged from 0.21 (CC2) to 2.52 (GS5) g/100 mL, with an average of 1.34 g/100 mL. The salinity of the samples ranged from 16.4 (JL8) to 35.2 (GG4) g/100 mL, with an average of 25.7 g/100 mL. The RS content ranged from 0.12 (GG4) to 1.46 (GG1) g/100 mL, with an average of 0.72 g/100 mL.

#### 3.1.2. Free Amino Acids

FAAs are important components of *ganjang* in terms of sensory response, contributing to its distinctive taste and aroma [47]; the composition of FAAs is a critical factor in determining its sensory quality [48,49]. The FAAs in *ganjang* result from enzymes breaking down the proteins in soybeans into smaller compounds, including amino acids, during the fermentation process [50]. In this study, 18 major amino acids were quantified in *ganjang* samples collected from different provinces (Table 2).

Glutamate was found to be the predominant component in almost all samples, followed by alanine, leucine, and lysine. The amounts of cysteine and methionine were observed to be relatively low compared with the other amino acids in the samples. Overall trends are consistent with previous reports on amino acid composition in traditional Korean soy sauce [51,52,53]. As active-taste FAAs, glutamate and aspartate are importantly responsible for the umami taste of soy sauce [54], and the sum of the two FAAs was abundant in JL2, GS8, CC8, and JJ1. Glycine, lysine, proline, serine, and threonine contribute to sweetness, and the sum of those FAAs was highest in JL2, followed by GS8, GG2, JL3, and CC1. Alanine is also an important active-taste FAA that contributes to sweet and sour taste. Among the samples, GG4 (1925.1 mg/100 g) had the highest alanine content, twice as high as GS7 (737.3 mg/100 g) and GS8 (727.2 mg/100 g), which were closely followed by GG4. The presence of this particularly high alanine in GG4 might be due to the addition of several sources of amino acids, such as meat, fish, and tofu. A high alanine content has also been reported in marinated pork soy sauce [53]. Meanwhile, FAAs related to bitter taste in *ganjang* include arginine, histidine, isoleucine, leucine, methionine, phenylalanine, tyrosine, and valine, which were more highly found in GG4, JL2, GS8, and GS5 than other samples. These are also produced during the fermentation process and contribute to the overall sensory profile. The bitter taste is generally considered unpleasant in foods, but in the right concentrations, it can add depth and complexity to the overall flavor profile of soy sauce [54]. Therefore, they contribute to the complexity and balance of the flavor of soy sauce.

#### 3.1.3. Organic Acids

OAs are known to contribute to flavor, specifically sour, tangy, and umami, in fermented foods [55]. The levels of OAs can vary depending on the fermentation process, ingredients, and degree of maturity of the *ganjang* products [13]. As a result, Table 3 shows seven OAs identified in 36 *ganjang* products. 

Traditional fermented foods are often closely associated with the manufactured area, as they utilize locally available ingredients and microbes that are unique to that area [56,57]. However, there were no clear differences among production provinces in this study, which may be due to differences in manufacturing methods or ingredients for each product rather than regional characteristics. For details, lactate and acetate were the most abundant OAs found in almost all samples, whereas citrate, tartrate, malate, succinate, and formate were also present, but in relatively low quantities. There was considerable variation in the OA profiles among the samples, indicating differences in the production process or microbial composition of the *ganjang*. More specifically, lactate and acetate were metabolites dominated by lactic acid bacteria and acetic acid bacteria, respectively. Lactate was highest in GG2, followed by JL6, JL2, GG4, CC4, and high levels of acetate were found in GG4, GG1, and GW1. The GG4 sample had higher lactate and acetate contents. Lactate is a stable type of OA with strong acidity, but it activates a weak sour taste [58], implying that abundant lactate has a mild sour taste that can contribute to a soft and subtle flavor. Another finding was that some samples had relatively high concentrations of succinate (CC7, JL3, GS2, GG3, CC1, CC6, and CC2) and malate (JL3, CC1, GG1, GG5, and JL5). These OAs are responsible for umami taste and may play a role in the flavor of *ganjang*.

### 3.2. Internal Preference Mapping

IPM has been employed to gain a deeper understanding of consumer preference for *ganjang* samples and to identify different consumer segments based on their preferences [59]. In this study, PCA from the consumers’ overall acceptance was performed, followed by HCPC on the consumer space, resulting in the identification of each cluster, as shown in Figure 2. The first and second PC summarized the position of each segmentation and the reflected samples in the PCA plot (Figure 2a), demonstrating that the direction of each arrow represents a preference for each consumer with corresponding samples and the length of the vectors is proportional to the magnitude of variance (Figure 2b).

Overall, the consumer preference trends showed little segmentation, with most individual vectors pointing toward positive PC1, but it was segmented into three clusters by PC2 during the further HCPC. Moreover, these three clusters revealed significant differences in a few specific samples (Figure 2c). When comparing clusters, consumers in Cluster 1 (consisting of 52 consumers) preferred CC7 products, while they disliked JL4, GG2, and JL2 more than the other clusters. Consumers in Cluster 2 (comprising 27 consumers) preferred JL8 products much more than the other clusters. Overall, consumers in Clusters 2 and 3 (consisting of 23 consumers) had similar preferences for *ganjang products*, except for the CC6, GS2, JL8, and GS3 products. Preferences for CC6, GS2, and GS3 in Cluster 2 were similar to those in Cluster 1. Consumers in Cluster 3 (consisting of 23 consumers) favored CC6, JL4, CC5, GG2, and JL2 products more than those in Cluster 1. However, they had a much lower preference for GS2, JL8, and GS3 than consumers in Clusters 1 and 2.

### 3.3. External Preference Mapping

EPM provides valuable insights into how consumers perceive and prioritize different product attributes and identify key drivers of consumer preference. As a first step in EPM, PCA using the sensory profiles of 36 *ganjang* samples was performed; a PCA plot of sensory attributes and corresponding *ganjang* samples on the first two PCs is shown in Figure 3a. The sum of the first and second components explained 51.3% of the total variance. The first dimension (Dim1) was positively associated with ‘bitterness’, ‘astringent_M’, ‘bitter_AT’, ‘metallic_M’, and ‘biting_M’, and negatively associated with ‘sweetness’, ‘sweet_AT’, ‘sweat_O’, and ‘umami’. The second dimension (Dim2) was positively associated with ‘body_M’, ‘color_A’, and ‘burnt_O’, and negatively associated with ‘sour_O’, ‘alcohol_O’, and ‘alcohol_F’. Samples GG4, GS1, CC9, and CC2 were positively correlated with the first dimension, while GS8, CC1, and GW3 were negatively correlated. Regarding Dim2, GS5 was positively associated, while GS6, GS2, and CC7 were negatively associated. From this projection, Dim1 was correlated with ‘bitterness’, ‘astringent_M’, ‘bitter_AT’, ‘metallic_M’, and ‘biting_M’ as the main components of the positive vector, and ‘sweetness’, ‘sweet_AT’, ‘sweet_O’, and ‘umami’ as negatively based on the squared of the cosine value (Cos2) at >0.5. Regarding Dim2, ‘body_M’, ‘color_A’, and ‘burnt_O’ were major components of the positive side, and ‘sour_O’, ‘alcohol_O’, and ‘alcohol_F’ were major components of the negative side. The coordinated samples showed that GG4, GS1, CC9, and CC2 were highly correlated positively, and GS8, CC1, and GW3 were negatively correlated with Dim1 based on the Cos2 value (both >0.5). In Dim2, GS5 was highly associated with the (+) side of Dim2, while GS6, GS2, and CC7 were highly associated with the (−) side of Dim2.

The locations of the 36 *ganjang* samples in the contour plot of the predicted consumer satisfaction percentage and their prediction scores are shown in Figure 3b,c, respectively. The contour map indicates consumer preference, consisting of a series of contour lines and different degrees of color, from light coral to deep green. Each representation, along with the borderline, means a different level of satisfaction for a particular sensory attribute. In Figure 3c, JL5 and GW3 were highly appreciated by 90% of consumers, followed by GS8, JL3, JL7, CC1, and JJ1, which were appreciated by 80% of consumers. These products were correlated with sweet (odor, taste, and aftertaste) and umami (taste and aftertaste), which may imply drivers of liking. In contrast, only 10% of consumers appreciated GS1, CC2, and CC9, and these products were correlated with ‘biting_M’, ‘metallic_M’, ‘beany_F’, ‘bitterness’, ‘bitter_AT’, and ‘astringent_M’, which may imply drivers of disliking.

Figure 3d shows the mean scores of consumer acceptance for each sample and the results of the ANOVA among the samples. CC1 (5.89) was significantly higher, followed by JL5, GS8, and JJ1. The lowest score of acceptance was found in GS1 (3.01), followed by CC2, JL1, GG3, and GS6. The results showed a clear distinction between consumers’ liking and disliking of samples, but the samples from GG1 (5.28) to CC9 (4.01), which were neither liked nor disliked, did not show a significant difference.

### 3.4. Drivers of Liking

Figure 4a shows a PLS plot that illustrates the correlation map of physicochemical properties and sensory attributes with consumer acceptability. The X-axis datasets are physicochemical properties and sensory attributes, while the Y-axis dataset is consumer acceptability. The biplot shows that dimension 1 (t1) accounts for 76.7% of the total variance, while dimension 2 (t2) explains 12.2% of the total variance. The acceptance factors, including overall, appearance, odor, taste/flavor, and mouthfeel, are predominantly located on the positive side of t1, indicating that t1 captures the main factors that influence consumer acceptability. In Figure 4b, the variable identification coefficients (VIDs) indicate the importance of each predictor variable in explaining the variation in overall acceptability. VIDs were calculated by summing the squared correlations between each predictor variable and the PLS latent variables, weighted by the contribution of each latent variable to the prediction of the response variable [60,61]. Thus, higher VIDs indicate more significant predictors of overall acceptance. Based on the VIDs, sweetness was the most strongly related sensory attribute to overall acceptance, followed by ‘umami’, ‘sweet_O’, ‘sweet_AT’, and ‘umami_AT’. Among the FAAs, threonine, serine, proline, glutamate, aspartate, and lysine are positively associated with overall acceptance. Moreover, lactate and malate (of OAs) and acidity (of chemical properties) are also associated with overall acceptance. Meanwhile, ‘beany_F’, ‘bitterness’, ‘metallic_M’, ‘bitter_AT’, ‘beany_O’, ‘chemical_F’, ‘astringent_M’, ‘fermented_O’, and ‘biting_M’ were identified as negative drivers in the PLS model. In summary, the results from PLS-R analysis offer valuable insights into the sensory and physicochemical factors that contribute to acceptability, as well as the inter-relationship among these factors.

## 4. Discussion

The main objectives of the present study were to investigate the physicochemical characteristics and consumer acceptability of CQT *ganjang* samples produced from different provinces in Korea. Additionally, the underlying reasons for such ratings were studied by investigating the association between consumer acceptability and physicochemical properties, including FAA and OA profiles. 

The results of the proximate analysis and general characteristics show wide variation among *ganjang* samples. Notably, the 36 *ganjang* samples had already been certified in CQT, but there was a wide range of compositions and characteristics. In particular, lipid, TN, acidity, and RS showed large variability, as much as a 10-fold difference between samples, demonstrating that the certification system of CQT did not aim at guaranteeing physicochemical similarity or a specific parameter range. This information is useful for manufacturers to understand the overall composition and variability of CQT products produced in Korea. Meanwhile, common characteristics were not found in the *ganjang* samples from each producing region. As suggested in Appendix A in the Appendix A, there were no significant differences among producing regions for most properties. Furthermore, in the regional categorization of CQT ganjang, the confidence ellipses absolutely overlapped for regions (Appendix A), indicating that discrimination by province was not possible. In many cases, traditional fermented foods are known to be closely tied to regional features, reflecting the local geography, climate, available ingredients, and microbial communities present in the surrounding environment [62,63]. However, the composition and characteristics of the CQT *ganjang* samples used in this study might be far more influenced by individual *ganjang* producers than by regions. In addition, the use of various additives is commonly accepted in the production of *ganjang*, although the specific ingredients utilized can vary greatly based on regional traditions, personal preferences, and other factors [64]. This is no restriction on their use, except for determining the main source of soybean and salt, if additives have an established tradition. The use of various additives may consequently be affected.

To understand consumer preference for CQT *ganjang*, IPM was performed by creating individual preference space and consumer segments using PCA and cluster analysis. Taken overall, the first two PC explained 24.2% of total variance, which is relatively low. This could be caused by the insufficient number of consumers, as suggested by Kang [65], even though there are many factors involved. Therefore, it did not fully explain the total variations, but partially showed that CQT *ganjang* consumers across the board have similar consumer preference trends. Subsequently, cluster analysis demonstrated consumer segments with little difference, showing that consumers were strongly directed toward the positive side of PC1. This means that most consumers tend to prefer similar aspects of *ganjang*, and they had a common ideal point for CQT *ganjang*. For further HCPC, three preference segments based on PC2 were found, indicating that a few significant differences between samples polarized consumers’ likes and dislikes among the identified consumer segments. In the EPM, PCA space was established based on sensory profiles and fitting consumer acceptance data. Compared with the ANOVA result of the overall acceptance ratings, it contains considerable information on the relationship between a consumer’s sensory profile and their appreciation for *ganjang*. The information of sensory attributes can be useful for improving the eating quality of *ganjang* products for manufacturers and researchers.

The PLS-R results explain the drivers of liking *ganjang* by the VIDs of overall acceptance. As reflected by higher positive VIDs, ‘sweetness’, ‘umami’, ‘sweet_O’, ‘sweet_AT’, and ‘umami_AT’ had the strongest impact on consumer liking (>0.85). These sensory attributes resulted in agreement with the high-appreciation space in the EPM. Subsequently, several AAs, such as threonine, serine, proline, glutamate, aspartate, and lysine, were added (>0.54). In previous studies, these factors were described as the drivers of liking soy sauce and the foods prepared with it [14,15,66], Moreover, AAs were mentioned in soybean paste (*doenjang*), which has the same product origin as soy sauce [25,67].

Interestingly, the current study noted that acidity and OAs, such as lactate and malate, as well as AAs, were also associated with consumer acceptability (>0.52). These chemical properties are generally associated with sourness, but they were less relevant to ‘sourness’, ‘sour_O’, and ‘sour_AT’ in this study. It can be assumed that this is due to any metabolite accompanying fermentation products that release lactate and malate being indicators primarily of the activity of lactic acid bacteria. Further study is worth discussing on acidity related to microbial communities. Meanwhile, the negative high VIDs were constructed mainly of sensory attributes that in order of increasing absolute value ‘beany_F’, ‘bitterness’, ‘metallic_M’, ‘bitter_AT’, ‘beany_O’, ‘chemical_F’, ‘astringent_M’, ‘fermentated_O’, and ‘biting_M’. Most of the attributes were typical negative aspects of foods, but the descriptions related to fermentation were also found to be less associated with acceptance. Jeon [15], Lee, Chung, and Kim [17], elucidated a more intense fermented fish flavor found only in traditional *ganjang* compared with a mass product (e.g., brewed or acid-hydrolyzed soy sauce). In this study, a strong fermented flavor might have been perceived negatively by some consumers, such as those who are used to mass-produced goods.

The overall findings have practical implications for CQT *ganjang* manufacturers that aspire to enhance desirable flavor characteristics while minimizing undesirable attributes, and may be a viable approach to improve consumer acceptance. Moreover, the results have important implications for the food industry, as they can contribute to the development of strategies to develop and optimize traditional food products.

## 5. Conclusions

This study provides valuable information on the physicochemical characteristics and consumer acceptability of CQT *ganjang* samples originating from various provinces in Korea. The results showed wide variation among *ganjang* samples in terms of their composition and sensory characteristics. However, common characteristics were not found in *ganjang* samples within each region, indicating that individual *ganjang* producers may have a more significant influence on *ganjang* composition than the producing regions of CQT *ganjang* samples. The study also identified the drivers of liking *ganjang*, with ‘sweetness’, ‘umami’, ‘sweet_O’, ‘sweet_AT’, and ‘umami_AT’ having the strongest impact on consumer liking, while attributes such as ‘bitterness’ and strong ‘fermented flavor’ were generally perceived negatively. The results of this study have practical implications for CQT ganjang manufacturers that aspire to enhance desirable flavor characteristics, minimize undesirable attributes, and contribute to the development of strategies for optimizing traditional food products.

## Figures and Tables

**Figure 1 foods-12-02361-f001:**
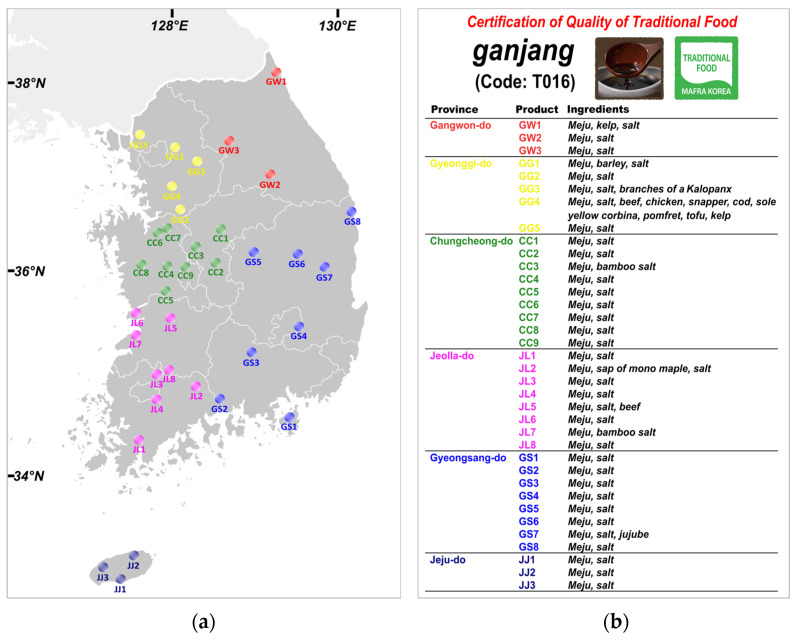
Schematic product information: (**a**) the provinces and locations where the products were produced; (**b**) ingredients for each *ganjang* sample with the CQT.

**Figure 2 foods-12-02361-f002:**
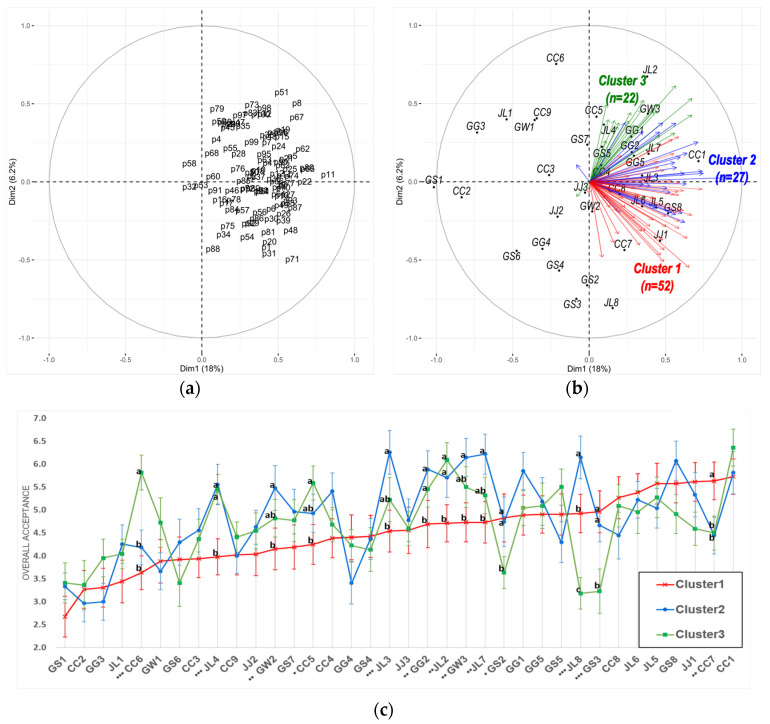
Internal preference map obtained from a PCA of consumers’ liking scores: (**a**) first two PC of consumers’ liking scores, (**b**) PCA biplot of individual liking scores colored according to the cluster corresponding to the samples, and (**c**) differences in averaged overall acceptance scores among the three consumer clusters. *, ** and *** represent a significant difference among the samples at *p* < 0.05, 0.01 and 0.001, respectively; different superscripts represent significant differences by Fisher’s LSD at *p* < 0.05.

**Figure 3 foods-12-02361-f003:**
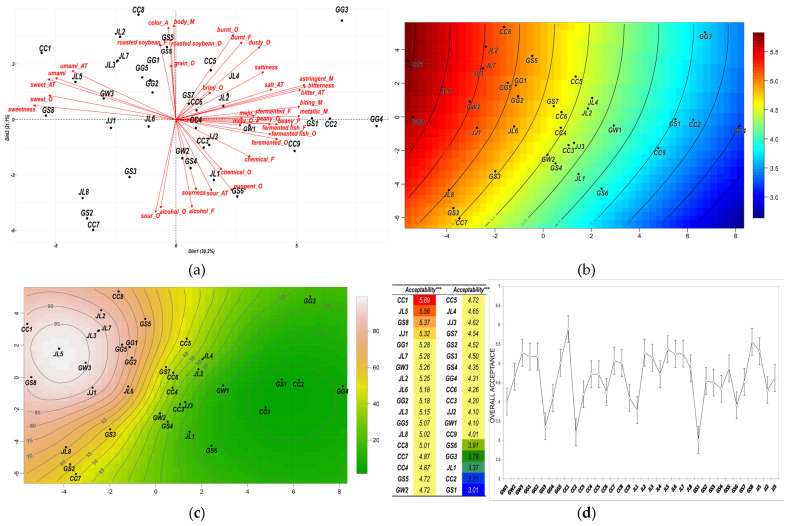
External preference maps based on the results of the PCA: (**a**) PCA biplot generated from sensory attributes corresponding to the samples (the red arrows represent sensory attributes; the dots represent samples. _A: appearance, _O: odor, _F: flavor, _AT: aftertaste); (**b**) predicted score map of *ganjang* samples; (**c**) contour preference map of *ganjang* samples corresponding to the external map generated by LOESS through GAM; and (**d**) mean scores of overall acceptance by consumers (*** represent a significant difference among the samples at *p* < 0.001; the same color represents no significant difference among the samples according to the SNK multiple comparison test, at *p* < 0.05).

**Figure 4 foods-12-02361-f004:**
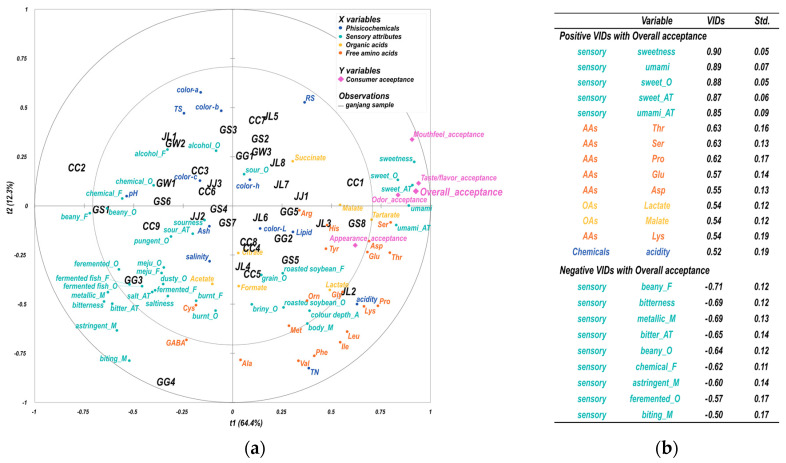
Loading plot generated by PLS-R: (**a**) correlation map by PLS-R of acceptance data as dependent variables (Xs), sensory attributes, and physicochemical data (Ys) as explanatory variables (_A: appearance, _O: odor, _F: flavor, _M: mouthfeel, _AT: aftertaste); and (**b**) positive and negative VIDs corresponding to overall acceptance in order of increasing absolute value.

**Table 1 foods-12-02361-t001:** Physicochemical properties ^1^ of 36 *ganjang* products collected from various provinces.

Product	TS ***^,2^ (%, *w*/*w*)	Ash *** (%, *w*/*w*)	Lipid *** (%, *w*/*w*)	TN *** (%, *w*/*w*)	Color-L ***^,3^	Color-a ***	Color-b ***	Color-c ***	Color h ***	pH ***	Acidity *** (g/100 mL)	Salinity *** (g/100 mL)	RS *** (g/100 mL)
GW1	31.5 ± 0.0 ^m,3^	24.3 ± 0.0 ^ef^	0.39 ± 0.04 ^fghij^	0.56 ± 0.00 ^E^	64.6 ± 4.1 ^a^	1.13 ± 0.14 ^d^	0.30 ± 0.63 ^e^	3.72 ± 0.07 ^l^	36.6 ± 0.4 ^lmn^	5.42 ± 0.04 ^k^	0.93 ± 0.05 ^jkl^	27.2 ± 0.7 ^abcde^	0.82 ± 0.02 ^l^
GW2	26.7 ± 0.2 ^s^	20.5 ± 0.0 ^m^	0.22 ± 0.06 ^kl^	0.46 ± 0.00 ^G^	68.2 ± 1.0 ^a^	1.55 ± 0.13 ^cd^	0.05 ± 0.20 ^e^	4.64 ± 0.02 ^h^	35.2 ± 0.2 ^nop^	6.39 ± 0.02 ^c^	0.42 ± 0.10 ^m^	25.3 ± 1.7 ^bcdefg^	0.35 ± 0.01 ^r^
GW3	25.4 ± 0.0 ^t^	17.6 ± 0.0 ^s^	0.34 ± 0.03 ^ijk^	0.88 ± 0.00 ^q^	69.4 ± 0.5 ^a^	4.18 ± 5.56 ^bcd^	−0.27 ± 0.29 ^e^	5.56 ± 0.12 ^k^	34.1 ± 1.6 ^nop^	5.00 ± 0.02 ^o^	1.17 ± 0.09 ^hijkl^	20.1 ± 1.6 ^fghi^	0.94 ± 0.01 ^ij^
GG1	34.7 ± 0.2 ^i^	24.2 ± 0.0 ^f^	0.28 ± 0.02 ^jkl^	0.80 ± 0.00 ^u^	50.5 ± 18.4 ^b^	3.85 ± 6.04 ^bcd^	2.51 ± 3.37 ^bc^	6.43 ± 0.06 ^rst^	33.0 ± 0.6 ^d^	5.23 ± 0.01 ^m^	1.23 ± 0.05 ^ghijk^	28.6 ± 2.3 ^abcd^	1.46 ± 0.06 ^a^
GG2	47.9 ± 0.2 ^c^	24.4 ± 0.0 ^ef^	0.51 ± 0.03 ^cdefg^	1.24 ± 0.00 ^h^	70.3 ± 0.0 ^a^	0.07 ± 0.02 ^d^	−0.82 ± 0.03 ^e^	5.72 ± 0.15 ^st^	33.7 ± 0.5 ^c^	5.58 ± 0.01 ^ij^	1.53 ± 0.32 ^efg^	31.3 ± 2.5 ^abc^	0.32 ± 0.01 ^rs^
GG3	34.3 ± 0.1 ^i^	25.1 ± 0.0 ^c^	0.20 ± 0.02 ^lm^	0.87 ± 0.00 ^r^	69.3 ± 0.5 ^a^	0.05 ± 0.02 ^d^	−0.75 ± 0.31 ^e^	4.94 ± 0.20 ^st^	33.3 ± 1.7 ^b^	5.34 ± 0.02 ^l^	1.11 ± 0.05 ^hijkl^	30.5 ± 1.7 ^abc^	0.20 ± 0.01 ^uv^
GG4	40.4 ± 0.1 ^d^	20.9 ± 0.0 ^l^	0.74 ± 0.02 ^a^	2.40 ± 0.00 ^a^	70.4 ± 1.4 ^a^	0.24 ± 0.01 ^d^	−0.83 ± 0.07 ^e^	4.20 ± 0.21 ^st^	33.8 ± 1.9 ^c^	5.72 ± 0.03 ^h^	1.86 ± 0.05 ^cd^	32.9 ± 3.3 ^a^	0.12 ± 0.02 ^w^
GG5	35.6 ± 0.0 ^h^	23.3 ± 0.0 ^h^	0.35 ± 0.07 ^hijk^	1.04 ± 0.01 ^m^	72.1 ± 2.9 ^a^	0.26 ± 0.10 ^d^	−0.71 ± 0.22 ^e^	3.12 ± 0.24 ^pqr^	44.7 ± 3.8 ^g^	5.05 ± 0.05 ^o^	1.62 ± 0.00 ^de^	31.5 ± 2.4 ^abc^	1.06 ± 0.00 ^fg^
CC1	35.2 ± 0.2 ^h^	24.1 ± 0.0 ^f^	0.40 ± 0.07 ^fghij^	0.99 ± 0.00 ^o^	71.8 ± 1.5 ^a^	−0.03 ± 0.02 ^d^	−0.64 ± 0.33 ^e^	2.13 ± 0.11 ^st^	56.6 ± 1.0 ^ab^	4.93 ± 0.04 ^p^	1.71 ± 0.16 ^de^	28.7 ± 1.7 ^abcd^	0.60 ± 0.01 ^o^
CC2	35.6 ± 0.3 ^h^	26.6 ± 0.7 ^a^	0.19 ± 0.03 ^lm^	0.26 ± 0.00 ^H^	70.1 ± 3.7 ^a^	3.41 ± 0.29 ^bcd^	2.07 ± 0.69 ^bcd^	1.13 ± 0.17 ^f^	67.5 ± 0.3 ^i^	6.88 ± 0.03 ^a^	0.21 ± 0.05 ^m^	31.7 ± 1.5 ^abc^	0.22 ± 0.03 ^tu^
CC3	29.2 ± 0.0 ^p^	22.2 ± 0.0 ^k^	0.52 ± 0.01 ^bcdef^	0.78 ± 0.00 ^w^	70.7 ± 1.6 ^a^	0.08 ± 0.08 ^d^	−0.77 ± 0.19 ^e^	1.16 ± 0.17 ^i^	70.8 ± 1.3 ^mno^	5.28 ± 0.02 ^lm^	1.02 ± 0.05 ^jkl^	26.4 ± 2.0 ^abcdef^	0.92 ± 0.02 ^j^
CC4	33.1 ± 0.3 ^j^	20.2 ± 0.0 ^n^	0.55 ± 0.06 ^bcde^	1.20 ± 0.00 ^i^	68.7 ± 2.6 ^a^	0.15 ± 0.07 ^d^	−0.60 ± 0.30 ^e^	1.12 ± 0.07 ^qrs^	74.3 ± 4.4 ^fg^	4.80 ± 0.01 ^q^	2.46 ± 0.27 ^a^	21.3 ± 1.2 ^efghi^	1.01 ± 0.03 ^h^
CC5	54.9 ± 0.6 ^a^	24.5 ± 0.0 ^e^	0.33 ± 0.02 ^ijk^	1.52 ± 0.00 ^b^	71.8 ± 1.9 ^a^	−0.27 ± 0.46 ^d^	0.47 ± 0.35 ^e^	1.01 ± 0.14 ^st^	77.7 ± 4.3 ^b^	5.30 ± 0.05 ^lm^	1.98 ± 0.16 ^bc^	26.8 ± 1.0 ^abcde^	0.63 ± 0.01 ^no^
CC6	25.4 ± 0.0 ^t^	17.3 ± 0.0 ^t^	0.43 ± 0.04 ^efghi^	0.79 ± 0.00 ^v^	70.2 ± 0.9 ^a^	0.51 ± 0.05 ^d^	−0.57 ± 0.03 ^e^	0.90 ± 0.14 ^n^	80.6 ± 2.0 ^jkl^	5.01 ± 0.04 ^o^	1.38 ± 0.10 ^efghi^	20.0 ± 2.4 ^fghi^	0.61 ± 0.02 ^no^
CC7	27.6 ± 0.4 ^r^	19.4 ± 0.0 ^op^	0.53 ± 0.14 ^bcde^	0.68 ± 0.00 ^A^	71.1 ± 0.0 ^a^	5.57 ± 0.13 ^b^	2.70 ± 0.29 ^bc^	0.93 ± 0.17 ^a^	82.2 ± 0.4 ^klmn^	5.07 ± 0.03 ^o^	1.41 ± 0.05 ^efghi^	20.8 ± 2.0 ^efghi^	1.42 ± 0.02 ^b^
CC8	30.4 ± 0.1 ^n^	20.5 ± 0.0 ^m^	0.38 ± 0.03 ^ghij^	1.24 ± 0.01 ^h^	71.6 ± 0.3 ^a^	0.37 ± 0.03 ^d^	−0.44 ± 0.13 ^e^	1.06 ± 0.15 ^rst^	84.6 ± 3.5 ^ef^	6.55 ± 0.08 ^b^	0.87 ± 0.10 ^l^	26.9 ± 3.1 ^abcde^	0.24 ± 0.01 ^t^
CC9	34.3 ± 0.1 ^i^	25.0 ± 0.0 ^c^	0.11 ± 0.01 ^mn^	0.84 ± 0.00 ^s^	71.2 ± 1.5 ^a^	0.43 ± 0.10 ^d^	−0.57 ± 0.18 ^e^	1.19 ± 0.10 ^no^	81.0 ± 2.7 ^ijk^	5.65 ± 0.03 ^i^	1.14 ± 0.05 ^hijkl^	32.2 ± 3.9 ^ab^	0.65 ± 0.02 ^mn^
JL1	30.5 ± 0.0 ^n^	23.6 ± 0.0 ^g^	0.08 ± 0.02 ^n^	0.52 ± 0.00 ^F^	72.0 ± 1.4 ^a^	3.50 ± 0.69 ^bcd^	1.24 ± 0.70 ^cde^	1.19 ± 0.17 ^d^	79.2 ± 0.6 ^jklm^	5.56 ± 0.03 ^j^	0.93 ± 0.05 ^jkl^	26.2 ± 2.8 ^abcdef^	1.32 ± 0.02 ^c^
JL2	38.6 ± 0.3 ^e^	22.0 ± 0.0 ^k^	0.19 ± 0.03 ^lm^	1.49 ± 0.00 ^c^	74.2 ± 1.1 ^a^	−0.07 ± 0.27 ^d^	0.38 ± 0.49 ^e^	1.05 ± 0.18 ^st^	75.6 ± 0.9 ^c^	5.01 ± 0.03 ^o^	2.13 ± 0.10 ^b^	26.0 ± 1.6 ^abcdef^	0.54 ± 0.01 ^p^
JL3	51.6 ± 0.2 ^b^	24.7 ± 0.0 ^d^	0.38 ± 0.07 ^ghij^	1.38 ± 0.00 ^e^	74.6 ± 4.7 ^a^	−0.13 ± 0.12 ^d^	0.05 ± 0.75 ^e^	1.21 ± 0.09 ^t^	69.9 ± 2.7 ^a^	5.56 ± 0.06 ^j^	1.71 ± 0.09 ^de^	28.2 ± 1.2 ^abcd^	1.06 ± 0.00 ^fg^
JL4	36.1 ± 0.2 ^g^	22.9 ± 0.0 ^j^	0.29 ± 0.04 ^jkl^	1.28 ± 0.00 ^g^	71.7 ± 5.3 ^a^	−0.02 ± 0.07 ^d^	−0.81 ± 0.28 ^e^	1.29 ± 0.14 ^st^	61.3 ± 0.6 ^ab^	5.02 ± 0.03 ^o^	2.13 ± 0.05 ^b^	26.5 ± 5.6 ^abcdef^	0.88 ± 0.01 ^k^
JL5	37.6 ± 0.1 ^f^	25.5 ± 0.0 ^b^	0.55 ± 0.06 ^bcde^	0.75 ± 0.00 ^x^	68.7 ± 1.0 ^a^	0.14 ± 0.05 ^d^	−0.42 ± 0.15 ^e^	1.44 ± 0.13 ^st^	53.9 ± 0.8 ^d^	5.82 ± 0.04 ^g^	1.17 ± 0.09 ^hijkl^	30.2 ± 3.4 ^abc^	1.12 ± 0.02 ^e^
JL6	30.2 ± 0.1 ^n^	18.6 ± 0.0 ^q^	0.46 ± 0.04 ^defghi^	1.02 ± 0.00 ^n^	68.7 ± 1.0 ^a^	0.13 ± 0.06 ^d^	−0.51 ± 0.20 ^e^	1.23 ± 0.21 ^pq^	62.8 ± 4.5 ^g^	5.84 ± 0.05 ^g^	1.14 ± 0.05 ^hijkl^	21.3 ± 1.4 ^efghi^	0.17 ± 0.01 ^v^
JL7	32.1 ± 0.3 ^l^	24.3 ± 0.0 ^ef^	0.81 ± 0.01 ^a^	0.73 ± 0.00 ^y^	75.8 ± 2.6 ^a^	−0.11 ± 0.01 ^d^	−0.33 ± 0.09 ^e^	1.14 ± 0.06 ^st^	74.6 ± 1.9 ^ab^	4.78 ± 0.02 ^q^	1.56 ± 0.05 ^def^	27.1 ± 1.1 ^abcde^	1.06 ± 0.01 ^fg^
JL8	23.8 ± 0.0 ^u^	15.6 ± 0.0 ^w^	0.57 ± 0.04 ^bcd^	0.75 ± 0.00 ^x^	75.6 ± 6.6 ^a^	0.72 ± 0.58 ^d^	−0.15 ± 0.25 ^e^	1.01 ± 0.10 ^j^	86.1 ± 1.8 ^op^	5.16 ± 0.02 ^n^	1.26 ± 0.09 ^fghij^	16.4 ± 0.7 ^i^	1.08 ± 0.02 ^f^
GS1	28.7 ± 0.0 ^q^	19.3 ± 0.0 ^p^	0.33 ± 0.04 ^ijk^	0.59 ± 0.00 ^D^	65.9 ± 5.9 ^a^	0.38 ± 0.16 ^d^	0.05 ± 0.80 ^e^	4.02 ± 0.05 ^pq^	73.0 ± 0.4 ^g^	6.41 ± 0.02 ^c^	0.45 ± 0.09 ^m^	21.4 ± 0.4 ^efghi^	0.19 ± 0.02 ^uv^
GS2	26.7 ± 0.0 ^s^	16.7 ± 0.1 ^u^	0.64 ± 0.06 ^b^	0.92 ± 0.00 ^p^	70.5 ± 0.2 ^a^	5.09 ± 0.08 ^bc^	3.13 ± 0.09 ^b^	6.99 ± 0.11 ^c^	59.1 ± 0.2 ^mno^	4.89 ± 0.02 ^p^	1.68 ± 0.14 ^de^	19.2 ± 3.4 ^ghi^	1.06 ± 0.03 ^fg^
GS3	37.8 ± 0.0 ^f^	25.0 ± 0.0 ^c^	0.24 ± 0.04 ^kl^	0.67 ± 0.00 ^B^	65.7 ± 0.9 ^a^	8.61 ± 0.09 ^a^	5.48 ± 0.03 ^a^	10.13 ± 0.08 ^b^	45.7 ± 0.4 ^klmn^	5.92 ± 0.05 ^f^	0.84 ± 0.05 ^l^	32.4 ± 4.3 ^ab^	1.28 ± 0.02 ^d^
GS4	25.6 ± 0.0 ^t^	16.1 ± 0.0 ^v^	0.40 ± 0.03 ^fghij^	1.04 ± 0.00 ^l^	65.6 ± 3.3 ^a^	4.15 ± 0.30 ^bcd^	2.13 ± 0.72 ^bcd^	8.64 ± 0.05 ^g^	41.6 ± 0.7 ^p^	5.44 ± 0.02 ^k^	1.26 ± 0.00 ^fghij^	17.8 ± 1.1 ^hi^	0.33 ± 0.01 ^rs^
GS5	36.1 ± 0.4 ^g^	22.7 ± 0.0 ^j^	0.47 ± 0.01 ^defgh^	1.35 ± 0.00 ^f^	63.8 ± 0.1 ^a^	0.17 ± 0.01 ^d^	0.73 ± 0.12 ^de^	7.30 ± 0.06 ^rst^	38.5 ± 2.3 ^b^	4.83 ± 0.02 ^q^	2.52 ± 0.09 ^a^	29.3 ± 1.0 ^abcd^	0.90 ± 0.03 ^jk^
GS6	26.7 ± 0.5 ^s^	17.4 ± 0.0 ^t^	0.61 ± 0.05 ^bc^	0.81 ± 0.00 ^t^	71.8 ± 0.0 ^a^	1.97 ± 0.02 ^cd^	0.55 ± 0.03 ^e^	5.82 ± 0.14 ^e^	34.9 ± 0.2 ^jklm^	6.33 ± 0.04 ^d^	0.90 ± 0.24 ^kl^	15.6 ± 4.2 ^i^	0.47 ± 0.01 ^q^
GS7	32.7 ± 0.1 ^k^	23.1 ± 0.0 ^i^	0.54 ± 0.01 ^bcde^	1.06 ± 0.00 ^k^	67.1 ± 0.0 ^a^	0.72 ± 0.94 ^d^	−0.01 ± 0.08 ^e^	4.38 ± 0.06 ^qrs^	44.7 ± 2.0 ^e^	6.03 ± 0.02 ^e^	0.90 ± 0.09 ^kl^	25.0 ± 1.0 ^cdefg^	0.13 ± 0.02 ^w^
GS8	35.2 ± 0.3 ^h^	19.6 ± 0.0 ^o^	0.55 ± 0.07 ^bcde^	1.44 ± 0.00 ^d^	64.8 ± 0.0 ^a^	1.08 ± 0.01 ^d^	0.51 ± 0.03 ^e^	2.81 ± 0.07 ^p^	51.7 ± 1.0 ^h^	5.60 ± 0.03 ^ij^	1.65 ± 0.14 ^de^	23.1 ± 1.8 ^defgh^	0.67 ± 0.02 ^m^
JJ1	29.8 ± 0.0 ^o^	18.3 ± 0.0 ^r^	0.52 ± 0.10 ^bcdef^	1.17 ± 0.00 ^j^	70.6 ± 0.0 ^a^	0.01 ± 0.02 ^d^	0.41 ± 0.02 ^e^	1.25 ± 0.08 ^st^	58.0 ± 1.2 ^b^	5.31 ± 0.04 ^lm^	1.56 ± 0.21 ^def^	20.0 ± 0.9 ^fghi^	1.02 ± 0.03 ^gh^
JJ2	31.4 ± 0.0 ^m^	24.4 ± 0.0 ^ef^	0.34 ± 0.06 ^ijk^	0.63 ± 0.00 ^C^	67.4 ± 2.2 ^a^	0.17 ± 0.02 ^d^	−0.34 ± 0.06 ^e^	1.11 ± 0.12 ^o^	65.8 ± 1.6 ^ij^	5.26 ± 0.09 ^lm^	1.44 ± 0.24 ^efgh^	29.0 ± 0.5 ^abcd^	0.30 ± 0.01 ^s^
JJ3	35.5 ± 0.3 ^h^	25.1 ± 0.0 ^c^	0.33 ± 0.03 ^ijk^	0.69 ± 0.00 ^z^	70.0 ± 0.0 ^a^	0.88 ± 0.00 ^d^	−0.25 ± 0.02 ^e^	1.07 ± 0.07 ^m^	74.8 ± 1.1 ^klmn^	5.26 ± 0.03 ^lm^	1.08 ± 0.09 ^ijkl^	31.0 ± 2.1 ^abc^	0.96 ± 0.01 ^i^

^1^ Mean values ± standard deviations of the three replications. ^2^ TS: total solid content; TN: total nitrogen content; RS: reducing sugar content; *** represents a significant difference among the samples at *p* < 0.001. ^3^ Different superscript letters within the same row represent significant differences at *p* < 0.05 in the SNK multiple comparison test.

**Table 2 foods-12-02361-t002:** Content of free amino acids ^1^ (mg/100 mL) in *ganjang* obtained from various provinces.

	Ala ***^,2^	Arg ***	Asp ***	Cys ***	GABA ***	Glu ***	Gly ***	His ***	Ile ***	Leu ***	Lys ***	Met ***	Orn ***	Phe ***	Pro ***	Ser ***	Thr ***	Tyr ***	Val ***
GW1	110.6 ± 0.0 ^opq,3^	36.8 ± 0.0 ^j^	66.3 ± 0.0 ^p^	n.d. ^4^	21.2 ± 0.0 ^nopq^	220.8 ± 0.1 ^p^	44.4 ± 0.0 ^mn^	31.3 ± 0.0 ^l^	68.8 ± 0.0 ^p^	123.7 ± 0.0 ^t^	138.7 ± 0.1 ^rs^	11.3 ± 0.0 ^s^	50.2 ± 0.0 ^r^	78.2 ± 0.1 ^st^	101.2 ± 0.0 ^no^	61.2 ± 0.2 ^s^	50.2 ± 0.2 ^q^	33.4 ± 0.1 ^op^	76.9 ± 0.1 ^q^
GW2	117.8 ± 0.0 ^op^	5.0 ± 0.0 ^v^	44.5 ± 0.1 ^q^	n.d.	26.0 ± 0.0 ^nopq^	178.7 ± 0.2 ^q^	49.7 ± 0.0 ^m^	10.7 ± 0.0 ^p^	61.0 ± 0.1 ^p^	104.8 ± 0.0 ^u^	152.7 ± 0.1 ^qr^	11.4 ± 0.0 ^s^	70.3 ± 0.0 ^no^	65.8 ± 0.0 ^u^	46.8 ± 0.0 ^t^	45.4 ± 0.0 ^u^	34.7 ± 0.1 ^s^	26.8 ± 0.0 ^r^	75.9 ± 0.1 ^q^
GW3	195.7 ± 0.2 ^lm^	2.5 ± 0.0 ^w^	75.3 ± 0.1 ^o^	n.d.	7.7 ± 0.0 ^pq^	389.6 ± 0.4 ^m^	74.7 ± 0.1 ^jk^	49.5 ± 0.0 ^h^	108.7 ± 0.1 ^mn^	178.7 ± 0.4 ^nop^	182.5 ± 0.2 ^p^	18.8 ± 0.0 ^q^	75.9 ± 0.1 ^lmn^	111.5 ± 0.1 ^pq^	113.3 ± 0.0 ^lm^	123.9 ± 0.2 ^m^	85.4 ± 0.1 ^mn^	90.8 ± 0.0 ^f^	112.4 ± 0.2 ^p^
GG1	114.2 ± 0.3 ^opq^	11.9 ± 0.0 ^rs^	98.1 ± 0.0 ^m^	n.d.	8.2 ± 0.0 ^pq^	306.7 ± 0.2 ^n^	43.3 ± 0.0 ^mn^	33.7 ± 0.0 ^k^	79.6 ± 0.4 ^o^	140.4 ± 0.2 ^s^	160.4 ± 0.0 ^q^	10.2 ± 0.0 ^st^	73.2 ± 0.0 ^mno^	87.5 ± 0.0 ^s^	97.4 ± 0.1 ^op^	92.6 ± 0.1 ^p^	64.6 ± 0.1 ^o^	71.2 ± 0.0 ^j^	83.7 ± 0.2 ^q^
GG2	465.8 ± 3.1 ^ef^	27.4 ± 0.2 ^m^	86.3 ± 0.7 ^n^	n.d.	450.0 ± 3.1 ^c^	293.9 ± 1.5 ^no^	141.0 ± 0.8 ^f^	50.0 ± 0.5 ^h^	216.9 ± 1.8 ^efg^	338.8 ± 3.1 ^efg^	408.6 ± 2.5 ^ef^	26.8 ± 0.1 ^lm^	192.5 ± 1.3 ^d^	198.8 ± 1.8 ^g^	212.2 ± 1.4 ^e^	211.1 ± 1.1 ^f^	190.9 ± 1.2 ^d^	114.1 ± 0.8 ^d^	234.8 ± 1.1 ^h^
GG3	196.5 ± 0.2 ^lm^	14.0 ± 0.0 ^pq^	116.8 ± 0.0 ^l^	n.d.	5.8 ± 0.0 ^pq^	373.5 ± 0.4 ^m^	66.9 ± 0.0 ^kl^	45.2 ± 0.1 ^i^	111.8 ± 0.1 ^mn^	174.0 ± 0.3 ^opq^	211.3 ± 0.2 ^o^	9.0 ± 0.0 ^tu^	132.3 ± 0.1 ^h^	105.4 ± 0.1 ^q^	90.8 ± 0.1 ^pq^	98.1 ± 0.1 ^o^	90.6 ± 0.1 ^m^	106.4 ± 0.0 ^e^	84.7 ± 0.2 ^q^
GG4	1925.1 ± 37.9 ^a^	8.6 ± 0.4 ^t^	160.2 ± 2.7 ^i^	31.7 ± 0.1 ^a^	1761.5 ± 27.7 ^a^	276.6 ± 4.5 ^o^	90.7 ± 1.6 ^i^	46.1 ± 0.6 ^i^	374.9 ± 6.1 ^a^	477.1 ± 8.1 ^b^	275.3 ± 4.4 ^l^	117.3 ± 2.0 ^a^	116.1 ± 1.9 ^i^	449.9 ± 7.7 ^a^	209.0 ± 4.4 ^e^	49.6 ± 0.8 ^tu^	61.1 ± 1.2 ^op^	31.1 ± 0.5 ^pq^	728.1 ± 15.2 ^a^
GG5	278.3 ± 0.3 ^j^	112.1 ± 0.2 ^b^	251.8 ± 0.1 ^e^	0.3 ± 0.0 ^j^	53.3 ± 0.2 ^lm^	508.4 ± 0.5 ^j^	110.7 ± 0.1 ^h^	76.6 ± 0.0 ^c^	225.3 ± 0.3 ^de^	345.1 ± 0.2 ^efg^	304.5 ± 0.5 ^jk^	28.2 ± 0.1 ^kl^	78.7 ± 0.1 ^lm^	177.9 ± 0.1 ^i^	282.0 ± 1.1 ^b^	202.5 ± 0.4 ^g^	181.7 ± 0.0 ^e^	70.0 ± 0.3 ^j^	233.7 ± 0.5 ^h^
CC1	248.0 ± 0.1 ^k^	10.5 ± 0.2 ^s^	325.9 ± 0.0 ^b^	0.1 ± 0.0 ^k^	12.0 ± 0.1 ^pq^	640.6 ± 0.2 ^g^	114.9 ± 0.1 ^h^	83.7 ± 0.3 ^b^	226.4 ± 0.1 ^de^	341.2 ± 0.4 ^efg^	336.6 ± 0.2 ^i^	26.5 ± 0.1 ^lm^	65.4 ± 0.0 ^op^	181.1 ± 0.6 ^i^	250.8 ± 1.7 ^c^	240.9 ± 0.3 ^c^	198.4 ± 0.4 ^c^	81.7 ± 0.1 ^g^	243.0 ± 0.2 ^h^
CC2	40.4 ± 0.0 ^r^	13.1 ± 0.0 ^qr^	13.4 ± 0.1 ^t^	n.d.	7.8 ± 0.0 ^pq^	60.3 ± 0.1 ^r^	10.7 ± 0.0 ^p^	3.9 ± 0.0 ^q^	16.8 ± 0.0 ^r^	29.1 ± 0.0 ^w^	26.3 ± 0.0 ^t^	1.2 ± 0.0 ^w^	12.0 ± 0.0 ^u^	23.6 ± 0.0 ^v^	14.3 ± 0.0 ^u^	8.9 ± 0.0 ^w^	9.7 ± 0.0 ^t^	10.2 ± 0.0 ^t^	21.1 ± 0.1 ^s^
CC3	196.3 ± 0.4 ^lm^	11.7 ± 0.0 ^rs^	33.2 ± 0.1 ^r^	1.0 ± 0.0 ^f^	40.1 ± 0.1 ^mn^	508.6 ± 0.2 ^j^	71.6 ± 0.1 ^jkl^	34.6 ± 0.0 ^k^	108.9 ± 0.4 ^mn^	193.8 ± 0.1 ^mn^	237.3 ± 0.4 ^n^	23.8 ± 0.1 ^no^	100.9 ± 0.1 ^j^	118.3 ± 0.2 ^op^	72.2 ± 0.2 ^s^	110.3 ± 0.2 ^n^	90.4 ± 0.0 ^m^	50.4 ± 0.1 ^m^	72.5 ± 0.1 ^qr^
CC4	464.8 ± 1.6 ^ef^	55.0 ± 0.4 ^g^	82.8 ± 0.0 ^n^	n.d.	536.5 ± 4.0 ^b^	383.7 ± 1.6 ^m^	143.1 ± 0.6 ^f^	48.5 ± 0.4 ^h^	221.0 ± 0.7 ^defg^	388.4 ± 1.1 ^c^	416.7 ± 2.0 ^de^	56.9 ± 0.3 ^d^	90.7 ± 0.4 ^k^	161.9 ± 0.6 ^j^	197.6 ± 1.5 ^f^	88.8 ± 0.4 ^p^	151.0 ± 0.1 ^h^	19.7 ± 0.2 ^s^	238.8 ± 1.2 ^gh^
CC5	431.0 ± 21.1 ^gh^	30.4 ± 1.1 ^l^	155.4 ± 6.9 ^ij^	n.d.	324.0 ± 16.1 ^g^	472.4 ± 22.4 ^k^	147.3 ± 7.4 ^f^	28.5 ± 1.7 ^m^	213.0 ± 11.1 ^fgh^	356.7 ± 17.1 ^de^	435.4 ± 22.1 ^cd^	25.2 ± 1.1 ^mn^	153.1 ± 8.0 ^f^	228.2 ± 10.1 ^e^	208.0 ± 7.6 ^e^	114.4 ± 5.5 ^n^	162.5 ± 7.7 ^g^	77.9 ± 3.8 ^hi^	263.1 ± 12.9 ^e^
CC6	175.1 ± 0.2 ^mn^	8.0 ± 0.0 ^t^	112.2 ± 0.3 ^l^	n.d.	11.3 ± 0.0 ^pq^	467.0 ± 1.3 ^k^	63.7 ± 0.2 ^l^	60.6 ± 0.1 ^e^	130.2 ± 0.1 ^l^	219.1 ± 0.1 ^l^	237.4 ± 0.2 ^n^	26.6 ± 0.0 ^lm^	82.1 ± 0.0 ^l^	139.4 ± 0.1 ^lm^	107.3 ± 0.4 ^mn^	130.8 ± 0.3 ^kl^	82.7 ± 0.1 ^n^	12.1 ± 0.0 ^t^	87.4 ± 0.9 ^q^
CC7	159.6 ± 0.0 ^n^	5.7 ± 0.0 ^uv^	195.4 ± 0.8 ^g^	6.7 ± 0.3 ^b^	11.2 ± 0.0 ^pq^	644.3 ± 0.9 ^g^	93.9 ± 0.1 ^i^	49.3 ± 0.0 ^h^	110.0 ± 0.0 ^mn^	163.9 ± 0.2 ^pqr^	239.9 ± 0.3 ^n^	22.9 ± 0.0 ^o^	20.5 ± 0.1 ^t^	95.9 ± 0.2 ^r^	111.8 ± 1.9 ^lm^	142.8 ± 0.0 ^j^	108.0 ± 0.3 ^k^	2.7 ± 0.0 ^u^	115.4 ± 0.2 ^op^
CC8	647.6 ± 6.3 ^c^	1.2 ± 0.0 ^wx^	21.3 ± 0.4 ^s^	1.2 ± 0.0 ^e^	13.2 ± 0.3 ^opq^	1100.8 ± 12.1 ^c^	201.6 ± 1.5 ^b^	12.0 ± 0.3 ^p^	139.1 ± 1.3 ^k^	201.2 ± 1.9 ^m^	446.0 ± 3.1 ^c^	50.4 ± 0.4 ^e^	343.8 ± 2.7 ^a^	131.3 ± 0.7 ^mn^	115.7 ± 0.9 ^lm^	4.8 ± 0.1 ^wx^	13.6 ± 0.1 ^t^	11.5 ± 0.3 ^t^	207.2 ± 1.6 ^ij^
CC9	293.6 ± 23.6 ^j^	43.0 ± 3.2 ^i^	47.5 ± 4.0 ^q^	0.9 ± 0.0 ^g^	209.7 ± 17.1 ^i^	365.9 ± 29.6 ^m^	92.9 ± 7.8 ^i^	28.6 ± 2.0 ^m^	149.2 ± 12.0 ^j^	238.9 ± 19.4 ^k^	261.9 ± 20.0 ^lm^	20.6 ± 1.7 ^p^	70.0 ± 5.4 ^no^	145.7 ± 11.9 ^kl^	120.1 ± 9.3 ^l^	51.0 ± 3.9 ^t^	57.7 ± 4.6 ^p^	42.3 ± 3.6 ^n^	150.7 ± 12.6 ^kl^
JL1	96.1 ± 0.1 ^pq^	54.0 ± 0.1 ^g^	69.6 ± 0.4 ^p^	0.3 ± 0.0 ^j^	38.7 ± 0.2 ^mn^	233.9 ± 0.4 ^p^	33.0 ± 0.1 ^o^	32.0 ± 0.1 ^l^	61.7 ± 0.4 ^p^	124.0 ± 0.0 ^t^	176.7 ± 0.3 ^p^	15.7 ± 0.1 ^r^	51.0 ± 0.1 ^r^	73.6 ± 0.2 ^tu^	84.5 ± 0.7 ^qr^	75.2 ± 0.1 ^r^	62.8 ± 0.2 ^op^	65.5 ± 0.1 ^k^	62.6 ± 0.1 ^r^
JL2	447.7 ± 3.4 ^fg^	33.4 ± 0.5 ^k^	318.3 ± 2.5 ^c^	n.d.	13.6 ± 0.2 ^opq^	1272.9 ± 9.5 ^a^	177.1 ± 1.4 ^c^	87.3 ± 0.8 ^a^	356.1 ± 2.2 ^b^	583.9 ± 4.2 ^a^	604.4 ± 5.2 ^a^	79.2 ± 0.5 ^c^	209.2 ± 1.6 ^c^	362.0 ± 3.1 ^b^	302.1 ± 3.3 ^a^	331.6 ± 2.6 ^a^	248.9 ± 2.3 ^a^	118.7 ± 1.1 ^c^	360.3 ± 2.0 ^c^
JL3	345.9 ± 6.1 ^i^	128.7 ± 3.1 ^a^	264.2 ± 4.2 ^d^	n.d.	28.1 ± 0.6 ^nop^	697.1 ± 13.3 ^f^	132.7 ± 2.4 ^g^	60.9 ± 1.1 ^e^	233.0 ± 4.0 ^d^	332.7 ± 5.5 ^fg^	369.6 ± 6.7 ^h^	7.4 ± 0.0 ^u^	105.1 ± 2.1 ^j^	193.6 ± 3.7 ^gh^	254.4 ± 3.8 ^c^	222.3 ± 4.0 ^e^	181.3 ± 3.4 ^e^	135.8 ± 2.1 ^b^	262.3 ± 4.9 ^e^
JL4	422.6 ± 18.8 ^h^	61.7 ± 2.3 ^f^	125.6 ± 5.7 ^k^	0.3 ± 0.0 ^j^	435.9 ± 18.6 ^d^	417.5 ± 19.4 ^l^	126.7 ± 5.6 ^g^	48.5 ± 2.1 ^h^	223.3 ± 9.3 ^def^	384.3 ± 16.1 ^c^	414.0 ± 17.5 ^e^	28.7 ± 1.2 ^kl^	187.1 ± 8.0 ^de^	227.6 ± 9.3 ^e^	232.9 ± 10.7 ^d^	113.5 ± 5.0 ^n^	171.0 ± 7.3 ^f^	121.1 ± 5.2 ^c^	251.4 ± 10.3 ^efg^
JL5	88.8 ± 0.1 ^q^	85.5 ± 0.2 ^d^	115.1 ± 0.1 ^l^	n.d.	14.5 ± 0.0 ^opq^	243.1 ± 0.3 ^p^	40.4 ± 0.0 ^n^	33.4 ± 0.0 ^k^	46.2 ± 0.0 ^q^	68.9 ± 0.1 ^v^	123.1 ± 0.2 ^s^	4.0 ± 0.0 ^v^	55.3 ± 0.1 ^qr^	26.2 ± 0.0 ^v^	78.4 ± 0.0 ^rs^	74.6 ± 0.1 ^r^	61.5 ± 0.0 ^op^	26.2 ± 0.0 ^r^	61.9 ± 0.0 ^r^
JL6	522.3 ± 23.8 ^d^	1.1 ± 0.0 ^wx^	151.8 ± 6.7 ^j^	0.2 ± 0.0 ^j^	94.7 ± 4.5 ^k^	775.3 ± 37.0 ^e^	153.7 ± 7.0 ^e^	27.4 ± 2.1 ^mn^	229.0 ± 10.5 ^de^	349.2 ± 16.3 ^def^	394.7 ± 20.2 ^fg^	41.2 ± 2.3 ^h^	138.3 ± 6.8 ^g^	208.7 ± 9.9 ^f^	225.5 ± 12.2 ^d^	25.8 ± 1.2 ^v^	52.6 ± 2.2 ^q^	35.3 ± 1.7 ^o^	265.8 ± 11.8 ^e^
JL7	290.3 ± 4.0 ^j^	4.6 ± 0.0 ^v^	115.2 ± 0.8 ^l^	0.1 ± 0.0 ^k^	13.3 ± 0.3 ^opq^	535.3 ± 6.6 ^ij^	94.0 ± 1.3 ^i^	51.8 ± 1.1 ^g^	120.2 ± 1.2 ^lm^	160.3 ± 1.8 ^qr^	229.3 ± 3.4 ^no^	8.5 ± 0.2 ^tu^	66.4 ± 0.9 ^op^	78.8 ± 0.6 ^st^	169.8 ± 2.8 ^h^	127.8 ± 1.9 ^lm^	114.7 ± 1.7 ^j^	52.2 ± 0.7 ^m^	144.6 ± 1.4 ^lm^
JL8	306.7 ± 9.1 ^j^	52.3 ± 1.5 ^h^	194.3 ± 5.7 ^g^	0.7 ± 0.0 ^h^	82.1 ± 2.5 ^k^	438.0 ± 12.0 ^l^	115.2 ± 3.5 ^h^	11.2 ± 0.4 ^p^	178.7 ± 5.5 ^i^	282.3 ± 7.8 ^i^	314.1 ± 8.6 ^jk^	43.0 ± 1.5 ^g^	159.3 ± 4.9 ^f^	152.5 ± 4.5 ^k^	150.0 ± 5.1 ^j^	174.9 ± 5.1 ^h^	140.6 ± 4.0 ^i^	61.4 ± 1.5 ^l^	197.8 ± 5.9 ^j^
GS1	300.8 ± 12.2 ^j^	0.4 ± 0.0 ^x^	23.3 ± 0.7 ^s^	n.d.	267.7 ± 11.0 ^h^	21.9 ± 0.8 ^s^	111.8 ± 4.7 ^h^	1.4 ± 0.0 ^r^	104.0 ± 4.0 ^n^	153.2 ± 5.9 ^rs^	122.3 ± 4.9 ^s^	14.9 ± 0.5 ^r^	24.8 ± 0.9 ^st^	76.3 ± 2.5 ^t^	17.4 ± 0.2 ^u^	1.3 ± 0.0 ^x^	1.9 ± 0.0 ^u^	4.3 ± 0.0 ^u^	127.2 ± 5.2 ^no^
GS2	252.4 ± 4.8 ^k^	81.4 ± 1.0 ^e^	29.8 ± 0.6 ^r^	0.8 ± 0.0 ^h^	338.8 ± 7.1 ^f^	38.3 ± 0.9 ^s^	72.8 ± 1.3 ^jk^	60.7 ± 1.3 ^e^	152.0 ± 2.6 ^j^	301.8 ± 5.7 ^h^	275.3 ± 5.4 ^l^	40.2 ± 0.7 ^h^	31.4 ± 0.3 ^s^	168.0 ± 3.2 ^j^	141.7 ± 3.4 ^k^	103.2 ± 1.5 ^o^	107.7 ± 2.0 ^k^	76.4 ± 1.4 ^hi^	157.3 ± 2.8 ^k^
GS3	130.2 ± 0.1 ^o^	13.9 ± 0.0 ^pq^	157.2 ± 0.0 ^ij^	6.5 ± 0.0 ^c^	3.9 ± 0.0 ^q^	542.9 ± 0.3 ^i^	69.7 ± 0.0 ^jkl^	63.5 ± 0.0 ^d^	121.6 ± 0.1 ^lm^	189.3 ± 0.0 ^mno^	227.4 ± 0.0 ^no^	27.6 ± 0.2 ^kl^	60.3 ± 0.0 ^pq^	106.0 ± 0.3 ^q^	79.7 ± 0.3 ^rs^	134.6 ± 0.1 ^k^	97.7 ± 0.2 ^l^	78.6 ± 0.0 ^h^	118.4 ± 0.1 ^op^
GS4	336.2 ± 5.5 ^i^	7.0 ± 0.0 ^tu^	262.1 ± 4.2 ^d^	0.5 ± 0.0 ^i^	59.6 ± 1.0 ^l^	517.5 ± 9.3 ^ij^	132.2 ± 2.3 ^g^	28.3 ± 0.4 ^m^	204.4 ± 3.5 ^h^	334.8 ± 5.5 ^fg^	322.4 ± 4.6 ^ij^	48.7 ± 0.7 ^f^	129.1 ± 2.2 ^h^	197.3 ± 3.3 ^g^	198.7 ± 0.3 ^f^	156.4 ± 2.6 ^i^	109.3 ± 1.8 ^k^	74.7 ± 1.3 ^i^	212.7 ± 3.2 ^i^
GS5	330.5 ± 2.2 ^i^	110.1 ± 0.5 ^c^	179.9 ± 1.3 ^h^	0.3 ± 0.0 ^j^	13.1 ± 0.1 ^j^	819.1 ± 5.5 ^opq^	116.4 ± 7.2 ^d^	40.9 ± 0.1 ^h^	233.3 ± 1.5 ^j^	393.2 ± 2.7 ^d^	382.0 ± 3.8 ^c^	51.9 ± 0.7 ^gh^	185.1 ± 1.3 ^e^	241.3 ± 1.7 ^e^	186.6 ± 0.3 ^d^	226.8 ± 1.7 ^g^	171.0 ± 1.2 ^d^	163.9 ± 0.9 ^f^	253.6 ± 1.6 ^ef^
GS6	486.4 ± 22.2 ^e^	1.9 ± 0.0 ^wx^	45.5 ± 2.2 ^q^	1.2 ± 0.0 ^e^	414.5 ± 19.6 ^e^	180.6 ± 8.2 ^e^	144.5 ± 6.6 ^q^	25.7 ± 1.3 ^f^	175.6 ± 8.5 ^n^	262.5 ± 11.9 ^i^	312.1 ± 15.2 ^j^	36.5 ± 1.8 ^jk^	30.0 ± 1.4 ^i^	139.4 ± 6.4 ^s^	109.4 ± 3.9 ^lm^	25.4 ± 0.8 ^lmn^	43.3 ± 1.7 ^v^	28.7 ± 1.3 ^r^	196.1 ± 9.4 ^j^
GS7	737.3 ± 11.2 ^b^	3.1 ± 0.0 ^w^	81.1 ± 2.0 ^n^	0.7 ± 0.0 ^h^	34.6 ± 0.5 ^h^	814.1 ± 13.5 ^no^	179.2 ± 3.1 ^d^	12.1 ± 0.1 ^c^	243.2 ± 3.8 ^p^	364.2 ± 6.2 ^c^	425.5 ± 7.0 ^d^	40.1 ± 1.1 ^de^	55.6 ± 1.0 ^h^	183.5 ± 3.3 ^qr^	166.2 ± 4.1 ^i^	103.5 ± 1.6 ^h^	42.3 ± 0.5 ^o^	5.5 ± 0.2 ^r^	276.9 ± 5.0 ^d^
GS8	727.2 ± 31.2 ^b^	24.8 ± 0.3 ^n^	205.5 ± 4.0 ^f^	1.7 ± 0.0 ^d^	22.3 ± 1.2 ^d^	1222.4 ± 34.1 ^nopq^	223.5 ± 8.0 ^b^	26.5 ± 1.2 ^a^	376.2 ± 13.8 ^n^	572.4 ± 22.4 ^a^	530.3 ± 27.9 ^a^	96.2 ± 3.6 ^b^	204.6 ± 10.0 ^b^	328.6 ± 13.8 ^c^	246.5 ± 15.3 ^c^	260.1 ± 6.4 ^c^	207.3 ± 5.2 ^b^	68.9 ± 0.6 ^b^	410.6 ± 18.7 ^b^
JJ1	284.7 ± 10.7 ^j^	20.3 ± 0.4 ^o^	334.3 ± 12.0 ^a^	0.7 ± 0.0 ^h^	25.8 ± 1.0 ^h^	782.8 ± 27.5 ^nopq^	169.7 ± 6.1 ^e^	5.2 ± 0.2 ^d^	211.1 ± 7.8 ^q^	327.1 ± 11.6 ^gh^	377.9 ± 14.1 ^g^	33.8 ± 1.2 ^gh^	159.2 ± 5.8 ^j^	187.3 ± 7.2 ^f^	155.4 ± 7.5 ^hi^	243.9 ± 8.4 ^ij^	194.5 ± 6.9 ^c^	91.4 ± 4.0 ^cd^	236.3 ± 8.9 ^h^
JJ2	204.0 ± 7.9 ^l^	15.2 ± 0.3 ^p^	88.4 ± 3.3 ^n^	1.3 ± 0.0 ^e^	144.6 ± 5.3 ^e^	290.8 ± 11.0 ^j^	66.6 ± 2.4 ^no^	19.7 ± 0.4 ^kl^	118.1 ± 4.6 ^o^	202.9 ± 7.8 ^m^	294.9 ± 11.6 ^m^	29.3 ± 1.0 ^k^	253.0 ± 9.2 ^k^	125.3 ± 3.9 ^b^	87.7 ± 3.8 ^no^	80.8 ± 3.3 ^qr^	86.0 ± 3.5 ^q^	87.9 ± 3.6 ^mn^	140.7 ± 5.3 ^lm^
JJ3	207.3 ± 4.8 ^l^	33.7 ± 0.5 ^k^	129.7 ± 4.1 ^k^	1.0 ± 0.0 ^fg^	15.8 ± 0.3 ^fg^	586.1 ± 13.1 ^opq^	76.7 ± 2.6 ^h^	57.1 ± 1.1 ^j^	121.3 ± 3.0 ^f^	199.4 ± 4.8 ^lm^	247.1 ± 4.3 ^m^	33.4 ± 0.7 ^mn^	81.5 ± 1.6 ^j^	124.5 ± 2.9 ^l^	161.4 ± 4.5 ^no^	130.6 ± 2.9 ^hi^	104.7 ± 2.5 ^kl^	88.3 ± 2.2 ^k^	136.0 ± 3.0 ^mn^

^1^ Mean values ± standard deviations of the two replications. ^2^ Ala, alanine; Arg, arginine; Asp, aspartic acid; Cys, cysteine; GABA, 4-aminobutyric acid; Glu, glutamic acid; Gly, glycine; His, histidine; Ile, isoleucine; Leu, leucine; Lys, lysine; Met, methionine; Orn, ornithine; Phe, phenylalanine; Pro, proline; Ser, serine; Thr, threonine; Tyr, tyrosine; Val, valine; *** represents a significant difference among the samples at *p* < 0.001. ^3^ Different superscript letters within the same row represent significant differences at *p* < 0.05 in the SNK multiple comparison test. ^4^ n.d.: Not detected.

**Table 3 foods-12-02361-t003:** Contents of organic acids ^1^ (mg/100 mL) in *ganjang* obtained from different provinces.

	Citrate ***^,2^	Tartarate ***	Malate ***	Succinate ***	Lactate ***	Formate ***	Acetate ***
GW1	17.8 ± 0.5 ^efgh,3^	0.2 ± 0.0 ^p^	2.2 ± 0.0 ^lmno^	8.5 ± 0.3 ^fghi^	67.5 ± 0.2 ^l^	3.2 ± 0.0 ^cde^	101.7 ± 1.3 ^b^
GW2	3.5 ± 0.4 ^no^	5.0 ± 0.2 ^klmn^	n.d.	1.3 ± 0.0 ^i^	79.4 ± 2.1 ^l^	1.7 ± 0.1 ^de^	11.5 ± 0.0 ^no^
GW3	6.8 ± 0.3 ^lmn^	11.1 ± 0.3 ^ghi^	3.9 ± 0.5 ^klm^	23.8 ± 1.0 ^cde^	185.2 ± 5.0 ^hi^	1.9 ± 0.0 ^de^	56.5 ± 0.4 ^c^
GG1	12.0 ± 0.1 ^jk^	7.9 ± 0.1 ^ijk^	51.1 ± 0.5 ^c^	25.1 ± 0.1 ^cd^	223.3 ± 0.2 ^gh^	1.7 ± 0.2 ^de^	203.6 ± 0.0 ^a^
GG2	13.8 ± 0.9 ^ij^	1.8 ± 1.0 ^nop^	10.1 ± 2.9 ^i^	15.8 ± 1.7 ^ef^	620.4 ± 0.3 ^a^	6.3 ± 0.3 ^cde^	37.2 ± 0.7 ^e^
GG3	18.8 ± 0.3 ^efg^	7.0 ± 0.0 ^kl^	0.6 ± 0.0 ^no^	34.7 ± 3.3 ^b^	131.9 ± 3.0 ^jk^	5.8 ± 1.6 ^cde^	22.9 ± 0.9 ^jk^
GG4	16.0 ± 0.0 ^fghi^	8.6 ± 0.0 ^hijk^	5.9 ± 0.1 ^jk^	0.5 ± 0.0 ^i^	355.9 ± 19.1 ^d^	8.5 ± 0.3 ^bcd^	205.3 ± 0.7 ^a^
GG5	18.3 ± 0.4 ^efgh^	21.4 ± 0.1 ^d^	49.8 ± 0.2 ^c^	27.5 ± 1.0 ^bcd^	149.4 ± 9.8 ^ijk^	1.2 ± 0.1 ^de^	38.6 ± 1.6 ^e^
CC1	5.9 ± 0.1 ^mn^	33.9 ± 0.1 ^c^	58.2 ± 0.5 ^b^	33.4 ± 0.6 ^b^	236.5 ± 0.9 ^fg^	3.3 ± 0.0 ^cde^	32.3 ± 0.2 ^fg^
CC2	2.2 ± 0.1 ^op^	0.9 ± 0.0 ^op^	3.1 ± 0.1 ^lmno^	30.4 ± 1.5 ^bc^	23.0 ± 0.1 ^m^	0.3 ± 0.0 ^de^	27.5 ± 0.3 ^hi^
CC3	29.6 ± 0.3 ^c^	3.3 ± 1.3 ^mnop^	1.6 ± 0.6 ^mno^	2.4 ± 1.2 ^hi^	141.1 ± 36.9 ^ijk^	0.5 ± 0.5 ^de^	25.7 ± 0.6 ^hij^
CC4	4.4 ± 1.2 ^no^	12.1 ± 0.3 ^gh^	3.0 ± 1.5 ^lmno^	10.5 ± 1.0 ^fgh^	355.7 ± 11.2 ^d^	1.6 ± 0.4 ^de^	48.5 ± 0.1 ^d^
CC5	24.7 ± 2.5 ^d^	6.5 ± 0.5 ^klm^	10.3 ± 2.7 ^i^	10.9 ± 0.6 ^fgh^	328.0 ± 7.6 ^d^	8.1 ± 1.8 ^bcde^	49.6 ± 0.5 ^d^
CC6	7.9 ± 3.2 ^lm^	12.5 ± 2.0 ^g^	1.7 ± 0.4 ^mno^	32.0 ± 13.0 ^b^	160.2 ± 35.9 ^ijk^	0.8 ± 0.8 ^de^	39.3 ± 0.3 ^e^
CC7	13.4 ± 0.3 ^ij^	11.6 ± 0.2 ^gh^	5.3 ± 0.2 ^jkl^	44.4 ± 0.0 ^a^	279.4 ± 0.8 ^ef^	0.3 ± 0.1 ^de^	25.3 ± 0.2 ^hij^
CC8	49.4 ± 0.9 ^a^	10.7 ± 0.1 ^ghij^	10.3 ± 0.1 ^i^	0.5 ± 0.0 ^i^	286.1 ± 62.8 ^e^	14.7 ± 9.2 ^a^	32.6 ± 0.6 ^fg^
CC9	20.1 ± 0.8 ^ef^	2.4 ± 0.0 ^nop^	0.8 ± 0.0 ^mno^	2.9 ± 0.0 ^hi^	221.3 ± 33.5 ^gh^	8.2 ± 7.4 ^bcde^	35.0 ± 1.6 ^ef^
JL1	9.6 ± 2.2 ^kl^	3.2 ± 1.3 ^mnop^	n.d.	23.1 ± 0.1 ^cde^	46.4 ± 0.1 ^lm^	2.4 ± 1.1 ^de^	26.0 ± 0.6 ^hij^
JL2	12.0 ± 0.5 ^jk^	16.0 ± 0.1 ^f^	17.8 ± 0.2 ^f^	23.0 ± 0.1 ^cde^	410.0 ± 27.3 ^c^	3.9 ± 0.0 ^cde^	37.5 ± 1.8 ^e^
JL3	16.8 ± 0.3 ^fghi^	35.8 ± 0.6 ^bc^	70.0 ± 4.3 ^a^	43.7 ± 0.4 ^a^	217.6 ± 14.1 ^gh^	8.4 ± 2.1 ^bcd^	36.7 ± 0.8 ^e^
JL4	21.4 ± 1.2 ^e^	10.6 ± 3.3 ^ghij^	2.6 ± 0.0 ^lmno^	11.9 ± 0.1 ^fg^	176.5 ± 1.4 ^ij^	4.7 ± 0.9 ^cde^	39.4 ± 4.7 ^e^
JL5	14.6 ± 0.7 ^hij^	7.4 ± 0.3 ^jkl^	38.8 ± 1.1 ^d^	11.5 ± 0.1 ^fg^	336.6 ± 2.3 ^d^	2.7 ± 0.2 ^cde^	21.6 ± 4.8 ^jk^
JL6	50.2 ± 3.2 ^a^	18.7 ± 1.3 ^e^	13.4 ± 0.4 ^gh^	1.8 ± 0.0 ^i^	482.0 ± 55.1 ^b^	0.5 ± 0.5 ^de^	12.9 ± 0.9 ^mno^
JL7	26.4 ± 0.9 ^d^	36.7 ± 0.0 ^b^	12.3 ± 0.3 ^hi^	7.0 ± 0.4 ^ghi^	254.5 ± 1.4 ^efg^	7.6 ± 4.5 ^bcde^	15.8 ± 1.4 ^lm^
JL8	17.7 ± 2.5 ^efgh^	11.4 ± 3.9 ^gh^	9.9 ± 1.0 ^i^	10.7 ± 0.8 ^fgh^	181.3 ± 8.1 ^hij^	0.9 ± 0.1 ^de^	13.8 ± 0.2 ^mn^
GS1	19.1 ± 0.6 ^efg^	4.1 ± 0.1 ^lmno^	2.5 ± 0.1 ^lmno^	2.6 ± 0.1 ^hi^	124.0 ± 0.4 ^k^	10.2 ± 7.5 ^abc^	12.6 ± 2.2 ^mno^
GS2	8.5 ± 0.5 ^lm^	4.0 ± 2.5 ^lmno^	0.3 ± 0.0 ^no^	34.9 ± 0.4 ^b^	286.4 ± 0.2 ^e^	3.8 ± 1.6 ^cde^	19.7 ± 0.1 ^kl^
GS3	0.1 ± 0.0 ^p^	8.6 ± 4.0 ^hijk^	0.2 ± 0.3 ^o^	2.3 ± 0.1 ^hi^	14.1 ± 1.2 ^m^	4.7 ± 0.7 ^cde^	5.6 ± 0.4 ^p^
GS4	15.6 ± 1.3 ^ghij^	6.6 ± 0.3 ^klm^	3.5 ± 2.6 ^klmn^	12.9 ± 10.3 ^fg^	156.7 ± 4.1 ^ijk^	5.0 ± 2.7 ^cde^	22.2 ± 3.8 ^jk^
GS5	17.0 ± 2.5 ^fghi^	13.7 ± 0.3 ^fg^	25.2 ± 0.4 ^e^	21.6 ± 0.3 ^de^	234.0 ± 1.0 ^fg^	4.7 ± 0.3 ^cde^	17.8 ± 0.2 ^l^
GS6	50.1 ± 1.2 ^a^	5.1 ± 0.5 ^klmn^	6.2 ± 0.3 ^jk^	8.9 ± 8.4 ^fghi^	163.5 ± 3.0 ^ijk^	n.d.	21.8 ± 1.2 ^jk^
GS7	38.9 ± 0.7 ^b^	12.2 ± 0.2 ^gh^	6.7 ± 0.0 ^j^	5.7 ± 0.0 ^ghi^	276.9 ± 1.1 ^ef^	2.5 ± 0.5 ^cde^	9.7 ± 1.5 ^no^
GS8	25.7 ± 0.1 ^d^	46.8 ± 0.5 ^a^	10.5 ± 0.4 ^i^	21.5 ± 0.1 ^de^	246.7 ± 3.3 ^efg^	n.d.	29.2 ± 0.9 ^gh^
JJ1	25.7 ± 0.1 ^d^	46.8 ± 0.5 ^a^	10.5 ± 0.4 ^i^	21.5 ± 0.1 ^de^	246.7 ± 3.3 ^efg^	13.1 ± 0.4 ^ab^	23.3 ± 0.3 ^ijk^
JJ2	19.6 ± 5.7 ^efg^	12.3 ± 2.8 ^gh^	15.0 ± 1.4 ^g^	8.1 ± 0.4 ^fghi^	132.1 ± 0.8 ^jk^	n.d.	16.3 ± 0.2 ^lm^
JJ3	27.0 ± 0.8 ^d^	13.2 ± 1.4 ^g^	15.4 ± 1.6 ^g^	16.3 ± 0.4 ^ef^	132.3 ± 0.3 ^jk^	0.1 ± 0.1 ^e^	9.2 ± 6.0 ^o^

^1^ Mean values ± standard deviations of the three replications. ^2^ *** represents a significant difference among the samples at *p* < 0.001. ^3^ Different superscript letters within the same row represent significant differences at *p* < 0.05 in the SNK multiple comparison test; n.d.: not detected.

## Data Availability

The data presented in this study are available on request from the corresponding author.

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
