# Peer review of "Consumer Preference of Traditional Korean Soy Sauce (Ganjang) and Its Relationship with Sensory Attributes and Physicochemical Properties"

_foods, 2023, doi:10.3390/foods12122361_

Round 1

Reviewer 1 Report

In this study, the authors investigate the physicochemical characteristics, sensory attributes, and consumer acceptance of the traditional soy sauce (ganjang) produced in different provinces of Korea. This topic is very interesting, and the provide some useful information for the utilization and development of traditional foods. Some suggestions are as follows:

Title: sensory attributes are not included

2.3 In this study, the intensities of sensory attributes of ganjang samples were evaluated using an eight-point category scale by 101 subjects. In general, the trained assessors/sensory panel are used to do the quantitative sensory evaluation, for they have good repeatability, reproducibility and consistency. Therefore, the training and qualification proof of subjects in this study should be given.

Table1-3: the data in table are suggested as mean value±SD.

3.2 Fig. 2, the sum of the PC1 and PC2 only explained 24.2% of the total variance. So, how to determine the validity of this result? 

Author Response

Dear Reviewer 1

Thank you for valuable review comments on the article. We found the comments very helpful and constructive. We have revised the manuscript in response to your comments, and our individual responses and reflections on each comment, Please see the attachment. If you have any further corrective opinions or question, we are happy to accommodate them.

Sincerely,

Sang Sook, Kim

Reviewer 2 Report

The manuscript presents analysis of the  physicochemical properties of fermented ganjang samples produced from different Korean provinces in relation to  consumer acceptability 

Few comments 

Necessary to add information how consumers were find, selected or invited to study 

Table 1 will be more informative if "Chroma and hue will be calculated and added, as these characteristics more correlate with visual perception by human eyes. 

L217 'organoleptic' should be replaced by 'sensory'

L216. "FFA" should be replaced by FAA 

Titles of 3.1.3. OAs and 3.1.2. FAAs  and etc better replace by full text, not to use abbreviations

Author Response

Dear Reviewer 2

Thank you for valuable review comments on the article. We found the comments very helpful and constructive. We have revised the manuscript in response to your comments, and our individual responses and reflections on each comment, Please see the attachment. If you have any further corrective opinions or question, we are happy to accommodate them.

Sincerely,

Sang Sook, Kim
